# Frailty among people with multiple sclerosis who are wheelchair users

Tobia Zanotto[1,2]*, Laura A. Rice[2,3,4], Jacob J. Sosnoff[1,2]

**1** Department of Physical Therapy, Rehabilitation Science and Athletic Training, School of Health Professions, University of Kansas Medical Center, Kansas City, KS, United States of America, **2** Illinois Multiple Sclerosis Research Collaborative, Interdisciplinary Health Science Institute, University of Illinois at Urbana-Champaign, Urbana, IL, United States of America, **3** Department of Kinesiology and Community Health, University of Illinois at Urbana-Champaign, Urbana, IL, United States of America, **4** Center on Health, Aging and Disability, University of Illinois at Urbana-Champaign, Urbana, IL, United States of America

* tzanotto@kumc.edu

## Abstract

### Background

Frailty is a biological syndrome arising from cumulative declines across multiple physiologic systems. Although recent reports have described elevated frailty levels in people with multiple sclerosis (MS) with minimal to moderate disability, very little is known about frailty in individuals with severe disability. The objective of the current investigation was to evaluate frailty through the deficit accumulation model and to explore the relationship of frailty with MS clinical subtypes, disease duration and fall-history in wheelchair users living with MS.

### Materials and methods

Standard validated procedures were used to calculate a frailty index in 45 wheelchair and scooter users living with MS (median age = 60.0[16.0] years, 82.2% female, patient determined disease steps score = 7.0). Information on demographics, MS clinical subtypes, disease duration, and six-month fall-history were collected as part of a standardized medical survey.

### Results

The mean frailty index score was 0.54 (standard deviation = 0.13). Overall, 91.1% and 8.9% of participants met objective diagnostic criteria for severe and moderate frailty, respectively. A one-way ANOVA revealed no significant differences (F = 0.054, p = 0.948) in the frailty index among participants with relapsing-remitting MS, primary progressive, and secondary progressive MS. No relationship between frailty and disease duration (r = -0.058, p = 0.706) was found. A univariable negative binomial regression analysis revealed a significant association between frailty index scores and the number of falls experienced in the previous six months (IRR = 1.75, 95% CI [1.06–2.91], p = 0.030).

**Data Availability Statement:** All relevant data are within the manuscript and its Supporting Information files.

**Funding:** This work was supported by a Mentor-based Rehabilitation Research Post-doctoral fellow grant awarded to JS (MB-1807-31633) and by a

research grant awarded to LR from the National Multiple Sclerosis Society (RG-1701-26862). The funders had no role in study design, data collection and analysis, decision to publish, or preparation of the manuscript. There was no additional external funding received for this study.

**Competing interests:** The authors have declared that no competing interests exist.

## Conclusion

The current study suggests that individuals with MS with advanced disability also live with coexisting frailty and that the frailty index may be a valuable tool in evaluating fall-risk in wheelchair users living with MS. The significant overlap observed between severe disability and severe frailty highlights the emerging need to untangle this bi-directional relationship to identify appropriate therapeutic pathways in the MS population living with advanced disability.

## Introduction

Frailty is a biological syndrome characterized by decreased resistance to stressors, resulting from cumulative declines across multiple physiologic systems and leading to clinical adverse outcomes such as falls, hospitalizations and mortality [1]. While this condition has been studied extensively in geriatric populations [2, 3], very little is known about frailty among people living with multiple sclerosis (pwMS). In a recent UK Biobank study, Hanlon et al. [4] reported that pwMS had a 15-fold higher risk of being frail compared to age-matched non-MS individuals, and that MS was the top long-term disorder associated with frailty. Due to the growing availability of effective disease-modifying therapies, which has increased the life expectancy of pwMS [5], there is an emerging need to explore the impact of this biological syndrome on health-related outcomes in MS. Indeed, a few recent reports have started to explore the prevalence and characteristics associated with frailty in ambulatory pwMS [6, 7]. Nevertheless, there is a dearth of knowledge regarding the clinical implications of frailty in individuals with more advanced disability.

While there are significant overlaps between measures of disability and frailty [8], it has been reasonably theorized that cumulative disability may be one of the main drivers of frailty in MS [6]. People with elevated scores ($\geq$7.0) on the expanded disability status scale (EDSS) or patient determined disease steps (PDDS) utilize, almost exclusively, wheelchairs and/or other assistive devices to support mobility. The inability to ambulate independently can lead to a cascade of negative health outcomes, which could accelerate the onset and severity of frailty. In a recent study of 118 ambulatory persons with MS, we observed that approximately two thirds of pwMS (EDSS range = 1.0–6.0) were either moderately or severely frail [9]. This observation would seem to suggest that a very high proportion, if not the entirety, of pwMS who utilize wheelchairs or scooters as the main form of mobility may live with co-existing frailty. To date, however, there are no published reports of frailty among pwMS who use wheelchairs or scooters full-time, and it is unknown 1) which MS characteristics are associated with frailty and 2) whether a cumulative frailty index could predict paramount clinical adverse outcomes such as falls in this segment of the MS population [10]. Indeed, previous research has suggested that between 40% and 75% of individuals with MS who use wheelchairs or scooters fulltime experience at least one fall in any six-month period [10, 11]. Particularly, most of these falls seem to occur at home during transfers (e.g., in the bathroom) [10]. While the etiology of falls is multifactorial, it is plausible that elevated frailty levels may increase the risk of falls by impinging on the ability to transfer (e.g., through muscle weakness) [1].

Therefore, the objective of this study was to quantify frailty through the deficit accumulation model in pwMS with advanced disability (wheelchair or scooter users) and to explore its association with MS characteristics such as clinical subtypes, and disease duration. As a secondary objective, we aimed to explore the relationship between frailty and fall-history. We

hypothesized that higher frailty index scores would be associated with a higher number of falls experienced in the previous six months.

## Materials and methods

### Study design, setting and participants

A cross-sectional study design was utilized to explore the association between a frailty index and MS characteristics, as well as fall-history in a convenience sample of pwMS who use wheelchairs or scooters. Demographic and clinical characteristics, frailty-related measures, and history of falls (i.e., number of falls experienced in the previous six months) were collected during a single assessment. The study consisted of a secondary analysis of baseline data collected as part of a multi-center non-randomized clinical trial focusing on a personalized fall prevention intervention (Clinicaltrials.gov Identifier: NCT03705364) [12]. Participants had to meet the following inclusion criteria: a) aged 18 years or older, b) confirmed diagnosis of MS according to the revised McDonald's diagnostic criteria 2017 [13], c) patient determined disease steps (PDDS) score of 7.0 (i.e., use of a wheelchair as primary form of mobility), d) self-reported ability to transfer independently or with minimal/moderate assistance, e) self-report of at least one fall in the previous 12 months, f) able to understand written and spoken English. Participant exclusion criteria were: a) diagnosis of other neurological disorders (e.g., stroke, Parkinson's disease), b) MS exacerbation in the last 30 days, c) score $\geq 10$ on the short blessed test (a cognitive impairment screening tool) [14], d) inability to maintain an upright position for at least an hour, and e) pregnancy. The study was conducted in agreement with the ethical principles for medical research involving human subjects, as set forth by the world medical association declaration of Helsinki. All procedures were independently approved by the Institutional Review Boards at the University of Illinois at Urbana-Champaign (Project #18124), the University of Illinois at Chicago (Project #2017–1045), and the Shepherd Center in Atlanta (Project #733). All participants provided written informed consent prior to data collection.

### Measurements

Demographic and clinical characteristics of the study participants were collected by means of a standardized self-report medical survey. Participants completed the PDDS, as a common proxy measure of disability in pwMS. The PDDS evaluates MS disability based on motor and ambulatory dysfunction and has nine ordinal levels ranging from 0 (normal) to 8 (bedridden) [15]. In addition to the PDDS, participants completed the 54-item multiple sclerosis quality of life questionnaire [16], which was used to derive several frailty-related measures relating to global health, physical, psychosocial and sexual function. Participants were also administered the California verbal learning test, the symbol digit modalities test, and the brief visuospatial memory test, all of which are validated measures of cognition in MS [17, 18]. Additional questionnaires utilized to collect frailty-related information included the Community Participation Indicators (CPI) and the Spinal Cord Injury Falls Concern Scale (SCI-FCS) [19–21]. In addition, participants were asked to report the number of falls and fall-related characteristics (e.g., timing, locations, precipitating factors, fall-related injuries, etc.) experienced in the previous six months, as part of a standardized fall-history survey. Participants were classified as recurrent fallers if they reported two or more falls, or non-recurrent fallers if they reported one or no falls [22].

### Frailty assessment

We identified 30 health deficits (through the health-related instruments listed in the Measurements section above), which were used to calculate a frailty index by following validated

standard procedures [23]. A minimum of 30 health deficits has been recommended to achieve sufficient accuracy in predicting adverse outcomes [24]. According to the deficit accumulation guiding principles [23, 24], the health deficits were selected if: 1) they were potentially associated with health status, 2) they did not saturate too early with aging, 3) their prevalence generally increased with age, 4) they covered a range of systems. The exact operationalization of the frailty index is summarized in Table 1. Overall, the deficit items encompassed a wide range of systems including global health, physical function, cognition, sexual and psychosocial function. The variables (deficit items) included in the frailty index were coded on a 0–1 scale as per standard protocol [23]. A single researcher experienced in frailty assessments (TZ) performed all the coding to minimize inter-assessor variability. While the binary variables were recoded to indicate the absence or presence of the deficit, ordinal variables were recoded as rank scores ranging from '0' (deficit is absent) to '1' (deficit is maximally expressed) [24]. Continuous variables were also recoded on a 0–1 scale based on either relevant cut-points from the literature [20, 25], if available, or through the lowest quintile of distribution method [26]. The frailty index was then calculated by summing all deficit-related scores and dividing the sum by the total number of possible deficits. Participants with more than 5% of missing deficit items were removed from the analysis, as per standard recommendations [23].

## Statistical analysis

Statistical analyses were performed with SPSS, version 27.0 (IBM, Inc., Armonk, NY). The Shapiro-Wilk test was utilized to check whether data were normally distributed. Differences in frailty index among clinical MS subtypes (relapsing-remitting, primary progressive, secondary progressive) were explored using one-way ANOVA. The relationship of frailty index with age, disease duration, and duration of mobility aid use was examined through Pearson and Spearman correlation analyses, depending on normal distribution assumptions. The association between frailty index and history of falls was modeled through univariable negative binomial regression analysis. For this analysis, the frailty index was divided by two times its standard deviation to rescale the incidence rate ratio [27]. In addition, due to an extreme outlier in the number of falls reported by one participant (n = 300), we applied a 96% winsorization to the fall-history data. An Independent t-test comparing the frailty index scores in recurrent and non-recurrent fallers was conducted as sensitivity analysis. The level of statistical significance for interpretation of findings was set at p ≤ 0.05.

## Results

The data from 48 pwMS who used a wheelchair or scooter who completed the baseline study procedures were used for this secondary analysis. We excluded three participants (6.3%) who had more than 5% of missing deficit items. This resulted in a total number of 45 participants who were included in the final calculation of the frailty index. The sociodemographic and clinical characteristics of the study population are summarized in Table 2. The frailty index was normally distributed and ranged from 0.24 to 0.81 (Fig 1). Overall, participants had a mean frailty index score of 0.54 ± 0.13. According to established frailty index cut points [28], all participants were frail, with four (8.9%) and 41 (91.1%) subjects meeting objective criteria for moderate and severe frailty, respectively.

The one-way ANOVA performed to compare frailty index scores among clinical subtypes of MS revealed that there were not statistically significant differences (F = 0.054, p = 0.948) among participants with relapsing-remitting MS, primary progressive, and secondary progressive MS (Fig 2). As a sensitivity analysis, we compared the frailty index between participants with relapsing-remitting MS and with progressive (both primary and secondary) MS. An

**Table 1. Operational definition of frailty index: List of deficits included as variables.**

| Items | Source | Domain | Coding |
|---|---|---|---|
| 1. Self-rated health? | MSQOL-54 | Global health | Poor = 1; Fair = 0.75; Good = 0.5; Very good = 0.25; Excellent = 0 |
| 2. How has your health changed in the last year? | MSQOL-54 | Global health | Much worse = 1; Somewhat worse = 0.5; Better/Same = 0 |
| 3. Are you using a mobility aid? | Demographic survey | Function | Yes = 1; No = 0 |
| 4. Does anyone assist you in the performance of activities of daily living? | Demographic survey | Function | Yes = 1; No = 0 |
| 5. Are you worried or concerned you might fall? | Demographic survey | Function | Yes = 1; No = 0 |
| 6. Cut down on usual activity in the past month? | MSQOL-54 | Function | Yes = 1; No = 0 |
| 7. Difficulty performing work/other activities in the past month? | MSQOL-54 | Function | Yes = 1; No = 0 |
| 8. Does your health limit you in bathing/dressing? | MSQOL-54 | Function | Yes = 1; A little = 0.5; No = 0 |
| 9. Does your health limit you in lifting/carrying groceries? | MSQOL-54 | Function | Yes = 1; A little = 0.5; No = 0 |
| 10. Does your health limit you in climbing one flight of stairs? | MSQOL-54 | Function | Yes = 1; A little = 0.5; No = 0 |
| 11. Does your health limit you in walking one block? | MSQOL-54 | Function | Yes = 1; A little = 0.5; No = 0 |
| 12. Does your health limit you in bending, kneeling or stooping? | MSQOL-54 | Function | Yes = 1; A little = 0.5; No = 0 |
| 13. How much bodily pain have you had in the past month? | MSQOL-54 | Pain | Very severe = 1; Severe = 0.8; Moderate = 0.6; Mild = 0.4; Very mild = 0.2; None = 0 |
| 14. Did you feel tired in the past month? | MSQOL-54 | Energy | All of the time = 1; Most of the time = 0.8; A good bit of the time = 0.6; Some of the time = 0.4; A little = 0.2; None of the time = 0 |
| 15. Did you have a lot of energy in the past month? | MSQOL-54 | Energy | All of the time = 0; Most of the time = 0.2; A good bit of the time = 0.4; Some of the time = 0.6; A little = 0.8; None of the time = 1 |
| 16. Spinal Cord Injury Falls Concern Scale (SCI-FCS) score | SCI-FCS | Psychosocial | >24 = 1; ≤24 = 0 [20] |
| 17. Community Participation Ratio | CPI | Psychosocial | <0.61 = 1; ≥0.61 = 0 [25] |
| 18. Have you been happy in the past month? | MSQOL-54 | Psychosocial | All of the time = 0; Most of the time = 0.2; A good bit of the time = 0.4; Some of the time = 0.6; A little = 0.8; None of the time = 1 |
| 19. Have you felt downhearted and blue in the past month? | MSQOL-54 | Psychosocial | All of the time = 1; Most of the time = 0.8; A good bit of the time = 0.6; Some of the time = 0.4; A little = 0.2; None of the time = 0 |
| 20. Accomplished less than you would like as a result of emotional problems in the past month? | MSQOL-54 | Psychosocial | Yes = 1; No = 0 |
| 21. Satisfied with sexual function in the past month? | MSQOL-54 | Sexual | Very dissatisfied = 1; Somewhat dissatisfied = 0.5; Neither dissatisfied nor satisfied = 0 |
| 22. Difficulty getting/keeping erection (male) or inadequate lubrication (female) in the past month? | MSQOL-54 | Sexual | Very much a problem = 1; Somewhat of a problem = 0.5; Little of a problem = 0 |
| 23. CVLT-II | CVLT-II | Cognition | Total words recalled: above first quintile of distribution = 0; lowest quintile = 1 |
| 24. BVMT | BVMT | Cognition | Total BVMT score: above first quintile of distribution = 0; lowest quintile = 1 |
| 25. SDMT | SDMT | Cognition | Raw SDMT score: above first quintile of distribution = 0; lowest quintile = 1 |
| 26. Have you had troubles with memory in the past month? | MSQOL-54 | Cognition | All of the time = 1; Most of the time = 0.8; A good bit of the time = 0.6; Some of the time = 0.4; A little = 0.2; None of the time = 0 |
| 27. Difficulty concentrating or thinking in the past month? | MSQOL-54 | Cognition | All of the time = 1; Most of the time = 0.8; A good bit of the time = 0.6; Some of the time = 0.4; A little = 0.2; None of the time = 0 |
| 28. 25-foot walk performance | 25-foot walk test | Physical performance | Unable to perform without cane = 1; able to perform without cane = 0 |
| 29. Body mass index | Demographic survey | Comorbidity | <18.5, ≥30 = 1; >25, <30 = 0.5; ≥18.5, ≤25 = 0 |

*(Continued)*

**Table 1.** (Continued)

| Items | Source | Domain | Coding |
|---|---|---|---|
| 30. Problems with bowel or bladder function interfering with normal social activities? | MSQOL-54 | Comorbidity | Extremely = 1; Quite a bit = 0.75; Moderately = 0.5; A little bit = 0.25; Not at all = 0 |

**Abbreviations:** MSQOL-54: 54-item Multiple Sclerosis Quality of Life questionnaire; SCI-FCS: Spinal Cord Injury Falls Concern Scale; CPI: Community Participation Indicators; CVLT-II: California Verbal Learning Test; BVMT: Brief Visuospatial Memory Test; SDMT: Symbol Digit Modalities Test.

independent t-test revealed no statistically significant differences (F = 0.007, p = 0.776) between these two groups. The Pearson and Spearman correlation analyses did not reveal any relationship between frailty index scores and disease duration (r = -0.058, p = 0.706), or age (ρ = -0.113, p = 0.471). In an additional Spearman analysis, no correlation between frailty index and duration of mobility aid use was found (ρ = -0.085, p = 0.584).

Fig 3 shows the distribution of the number of falls reported as part of the six-month fall-history survey, after the winsorization. Overall, 62.2% of participants were classified as recurrent fallers as they reported falling at least twice. In univariable negative binomial regression analysis, the frailty index was associated with a greater number of falls reported in the previous six

**Table 2. Sociodemographic and clinical characteristics: Results are expressed as percentages for categorical variables and mean ± SD or median [IQR] for continuous variables.**

| Variables | Participants (n = 45) |
|---|---|
| Gender, female, n, (%) | 37 (82.2) |
| Age (years) | 60.0 [16.0] |
| BMI (kg/m$^2$) | 27.8 [7.3] |
| Ethnic background, n, (%) | |
| *African American* | 13 (28.9) |
| *Caucasian* | 30 (66.7) |
| *Hispanic* | 2 (4.4) |
| Marital status, n, (%) | |
| *Married* | 24 (53.3) |
| *Single* | 10 (22.2) |
| *Divorced* | 7 (15.6) |
| *Widow/er* | 4 (8.9) |
| MS clinical subtypes, n, (%) | |
| *Relapsing-remitting* | 18 (40.0) |
| *Primary progressive* | 9 (20.0) |
| *Secondary progressive* | 16 (35.6) |
| *Unsure* | 2 (4.4) |
| Disease duration (years) | 22.8 ± 11.3 |
| Type of mobility aid, n, (%) | |
| *Power wheelchair* | 29 (64.4) |
| *Manual wheelchair* | 8 (17.8) |
| *Scooter* | 8 (17.8) |
| Duration of mobility aid use (years) | 8.0 [8.8] |
| Recurrent falls, n, (%) | 28 (62.2) |

**Abbreviations:** SD standard deviation; IQR: interquartile range; BMI: body mass index; Recurrent falls: proportion of participants with at least two self-reported falls in the previous six months.

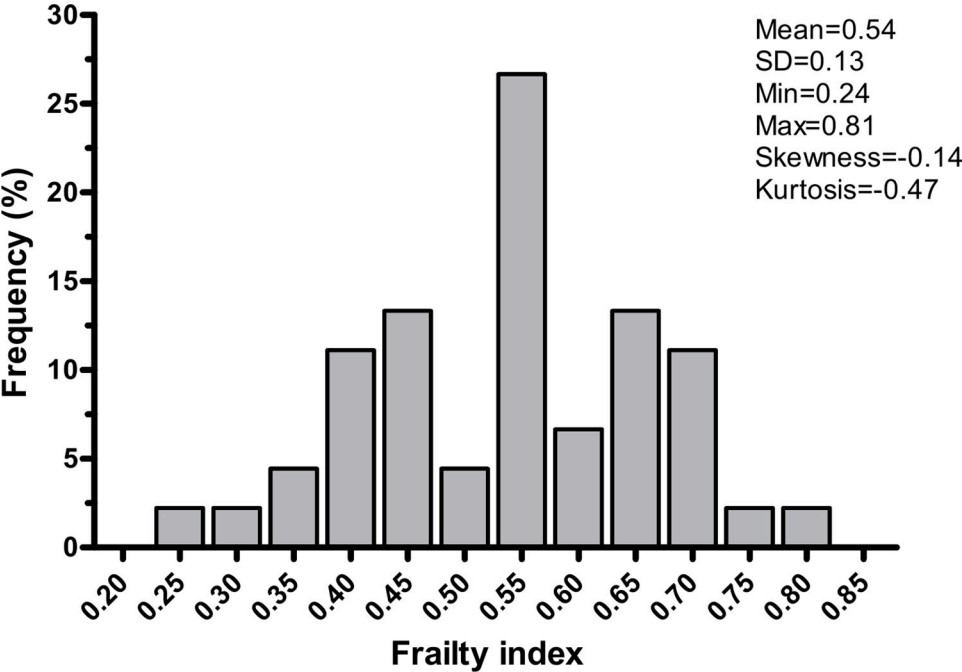

**Fig 1. Normal distribution of the frailty index in the study population.** SD = standard deviation.

months (IRR = 1.75, 95% CI [1.06–2.91], p = 0.030). As a sensitivity analysis, a further independent t-test did not reveal any statistically significant differences in frailty index between recurrent fallers and non-recurrent fallers (0.56 ± 0.11 vs 0.49 ± 0.14, p = 0.100).

## Discussion

In the current investigation, we aimed to quantify frailty through the deficit accumulation model in a population of pwMS with advanced disability (wheelchair or scooter users). The

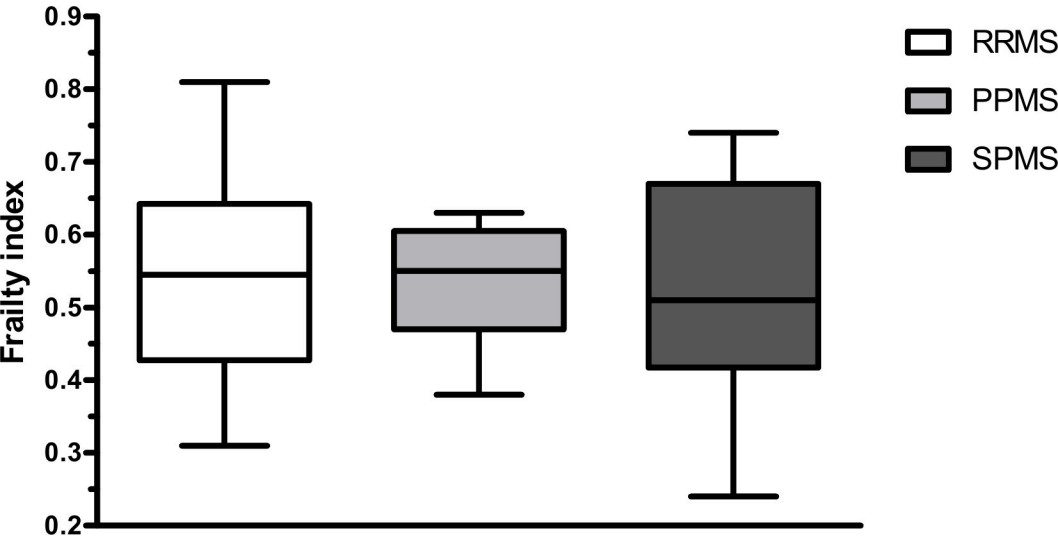

**Fig 2. Box plots of the frailty index across different MS clinical subtypes.** RRMS = relapsing-remitting multiple sclerosis, PPMS = primary progressive multiple sclerosis, SPMS = secondary progressive multiple sclerosis.

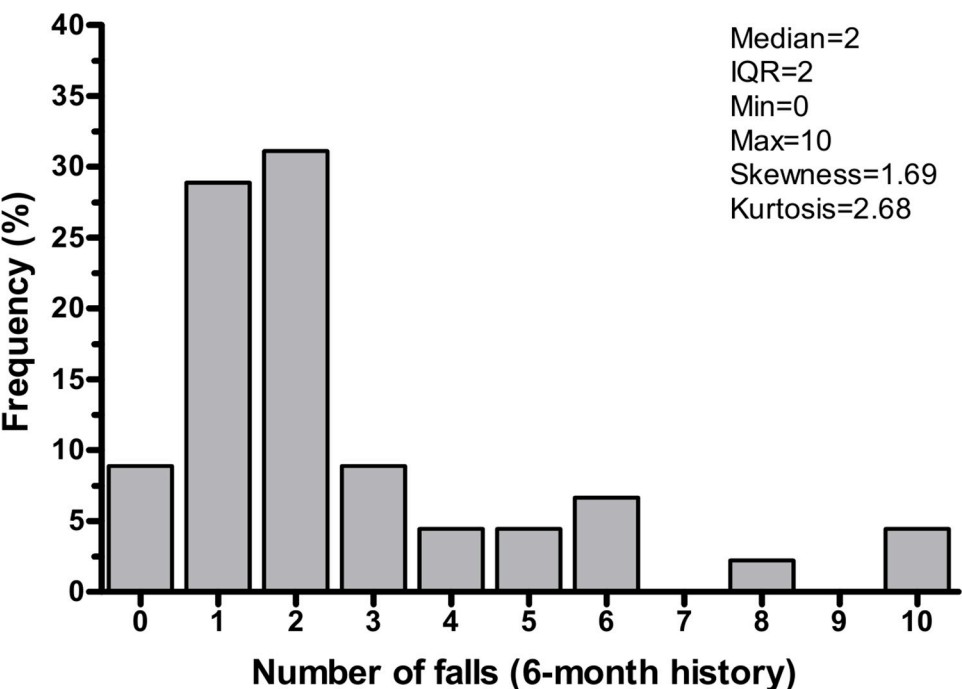

**Fig 3. Distribution of number of falls in the previous six months (fall-history).** IQR = interquartile range.

mean frailty index obtained through our operationalization (Table 1) strongly suggests that pwMS who utilize a wheelchair or scooter as the primary form of mobility are severely frail (Fig 1). As a secondary aim, we explored the relationship between the frailty index, MS characteristics (clinical subtypes and disease duration), and six-month fall-history. While no correlations between frailty and clinical subtypes of MS/disease duration were highlighted, the negative binomial regression analysis revealed a statistically significant association between frailty index scores and the number of falls experienced in the previous six months.

In a recent cross-sectional analysis, we provided initial evidence that cumulative frailty (i.e., frailty index approach) is a significant predictor of fall-history independently of disability levels, as assessed through the EDSS, and we postulated that frailty is a syndrome related to but independent of disability in pwMS who are able to ambulate independently (EDSS range: 1.0–6.0) [9]. In the current analysis, we sought to evaluate frailty in people at the higher end of the disability spectrum (PDDS = 7.0). Although it was not possible to examine the relationship between scores on the frailty index and the PDDS, due to the very limited range of disability included, it is nonetheless plausible to conclude that pwMS with severe disability also live with coexisting frailty. According to widely utilized cut points from the geriatric literature, frailty indexes > 0.36 are indicative of severe frailty [28]. More than 90% of participants in the current study met these objective criteria, while the remainder had scores reflecting moderate frailty (0.24–0.36). Importantly, none of the participants were non- or minimally-frail according to the frailty index literature [28, 29]. This observation strongly suggests that there may be a substantial overlap between severe disability and severe frailty in pwMS who use wheelchair or scooters. Previous research has proposed that disability may be one of the main drivers of frailty in pwMS [6]. In other words, the onset of MS-related disability may antecede the manifestation of frailty, a condition often referred to as secondary frailty [30], which differs in etiology from primary frailty (i.e., due to aging processes). In this respect, it should be noted that participants in the current study had a median age of 60 years and that no correlation between

the frailty index and age was found, which may reflect the premature onset of frailty of pwMS with advanced disability.

Although we did not include a control group in our investigation, the mean frailty index score of participants was 0.54 ± 0.13, which is considerably higher than values previously reported for pwMS affected by minimal to moderate disability (0.21 ± 0.12 and 0.32 ± 0.14) [6, 9]. While it is not possible to directly compare these frailty index scores, due to the different operationalizations employed, our findings provide compelling evidence of higher frailty levels in pwMS who use wheelchairs or scooters compared to individuals with more preserved ambulatory function. Interestingly, three participants (6.7% of our sample) exceeded a frailty index score of 0.7, which is commonly described as the upper physiological cut-off point, above which homeostasis reaches its limit and survival is severely compromised [31]. In addition, the frailty index was normally distributed rather than positively skewed, as it is often reported in geriatric populations [32]. This observation could reflect the fact that participants were quite similar in terms of disability levels, which may explain the relatively low heterogeneity of frailty index scores and may be once again construed as indirect evidence in support of the relationship between disability and frailty in pwMS.

Surprisingly, we did not find any relationship between frailty and MS clinical subtypes, disease duration or duration of mobility aid use. Recent reports have reasonably suggested that progressive MS may be the phenotype more frequently associated with frailty [6, 7]. However, it should be highlighted that both these investigations were conducted in individuals with lower disability, and the apparent discrepancy with our findings may be reconciled by the possibility that severe disability (PDDS = 7.0) and frailty may be too strongly associated [33], to observe the mediating effects of different clinical subtypes and disease duration in pwMS who use wheelchairs. On the other hand, the frailty index was significantly associated with fall-history, which indirectly suggests that the current definition of frailty (Table 1) may be valuable in predicting clinical adverse outcomes in people with advanced disability. This is consistent with findings from a recent study, in which a similar operationalization of the frailty index was a strong predictor of 12-month fall-history in ambulatory individuals with MS [9]. It should be noted that the variability of the frailty index scores in the current investigation, as quantified through the coefficient of variation metric, was considerably lower compared to our previous study (24% vs 44%), which is not surprising in light of the more homogeneous population included (i.e., wheelchair and scooter users only). Nevertheless, while there was no variation in the PDDS, our data clearly showed that there were more interindividual fluctuations in the frailty index. This observation seems to open up the possibility that, in pwMS who have reached the upper end of conventional measures of disability (e.g., PDDS, EDSS), the deficit accumulation model (i.e., frailty index approach) may be able to provide a more fine-grained measure to explore the risk of important adverse outcomes and/or response to intervention. While this is an intriguing hypothesis, further observational studies would be required to shed more light on the relationship between frailty and adverse outcomes in pwMS with severe disability.

## Study limitations

The current study is not without limitations. First of all, the lack of a control group could be seen as a study limitation in that we could not directly compare the frailty scores of individuals with advanced disability (wheelchair or scooter users) with those of individuals with minimal/moderate disability and/or healthy controls. Secondly, although 30 deficit items to operationalize the frailty index are generally reported to be sufficient to compute a reliable frailty index, it should be acknowledged that a greater number of items may provide stronger estimates of risk

[24]. In addition, findings from the current investigation should be carefully interpreted with the caveat that different conceptualizations (e.g., Fried's phenotype) and operationalizations of frailty exist and may yield different results. In this regard, one of the main criticisms of the frailty index approach is that it may fail to differentiate frailty from disability, as items portraying dependence in activities of daily living are often included in the operational definition [34]. Lastly, we should highlight that the sample size was relatively small, which may have increased the chances of committing a type II error.

## Conclusions

The current investigation revealed that pwMS who use wheelchairs or scooters were severely frail according to our operationalization of frailty. While no relationships between frailty and MS clinical subtypes/disease duration were identified, the frailty index was significantly associated with fall-history. This finding highlight the emerging need to evaluate the clinical/rehabilitative implications of frailty, as a potential corollary of disability, in this segment of the MS population. Future studies adopting different theoretical frameworks of frailty (e.g., physical frailty, comprehensive geriatric assessment) would be required to better understand the intersection/overlap between disability and frailty in pwMS living with advanced disability. Ultimately, this could lead to the development and implementation of tailored intervention strategies designed to more effectively target the rehabilitation needs of the MS community.

## Supporting information

**S1 Dataset.**
(XLSX)

## Acknowledgments

We would like to express our gratitude to the study participants for their time and contribution to this research project.

## Author Contributions

**Conceptualization:** Tobia Zanotto, Laura A. Rice, Jacob J. Sosnoff.

**Data curation:** Tobia Zanotto, Laura A. Rice.

**Formal analysis:** Tobia Zanotto.

**Funding acquisition:** Laura A. Rice, Jacob J. Sosnoff.

**Investigation:** Tobia Zanotto, Laura A. Rice.

**Methodology:** Tobia Zanotto, Jacob J. Sosnoff.

**Project administration:** Laura A. Rice, Jacob J. Sosnoff.

**Resources:** Laura A. Rice, Jacob J. Sosnoff.

**Visualization:** Tobia Zanotto.

**Writing – original draft:** Tobia Zanotto.

**Writing – review & editing:** Laura A. Rice, Jacob J. Sosnoff.

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
