## [Decision Letter · Decision Letter 0]

2 Jun 2022

PONE-D-21-38708Frailty among people with multiple sclerosis who are wheelchair usersPLOS ONE

Dear Dr. Zanotto,

Thank you for submitting your manuscript to PLOS ONE. After careful consideration, we feel that it has merit but does not fully meet PLOS ONE’s publication criteria as it currently stands. Therefore, we invite you to submit a revised version of the manuscript that addresses the points raised during the review process.

We look forward to receiving your revised manuscript.

Kind regards,

Sreeram V. Ramagopalan

Academic Editor

PLOS ONE

Journal Requirements:

2. Thank you for including your ethics statement:  "The study was conducted in agreement with the ethical principles for medical research involving human subjects, as set forth by the world medical association declaration of Helsinki. All procedures were independently approved by the Institutional Review Board at each study center. All participants provided written informed consent prior to data collection.".  

“This work was supported in part by a Mentor-based Rehabilitation Research Post-doctoral fellow grant awarded to JS (MB-1807-31633) and other funding awarded to RL (RG-1701-26862) from the National Multiple Sclerosis Society. The funders had no role in study design, data collection and analysis, decision to publish, or preparation of the manuscript.”

Reviewers' comments:

Reviewer's Responses to Questions

**Comments to the Author**

1. Is the manuscript technically sound, and do the data support the conclusions?

Reviewer #1: Yes

2. Has the statistical analysis been performed appropriately and rigorously? 

Reviewer #1: Yes

3. Have the authors made all data underlying the findings in their manuscript fully available?

Reviewer #1: Yes

4. Is the manuscript presented in an intelligible fashion and written in standard English?

Reviewer #1: Yes

5. Review Comments to the Author

Reviewer #1: Thank you for the opportunity to review this paper. It's written excellently and has robust conclusions aligned with the results and data.

My only feedback is that I find the connection between frailty and wheelchair use difficult to comprehend/ appreciate. If someone is using a wheelchair or scooter full time, isn't their chance of falling quite small- how are these patients falling over so much, is it in transition from the wheelchair/ scooter to another sitting state (e.g., to the sofa or bed?) I wonder if it's worth a mention in the intro somehow?

6. PLOS authors have the option to publish the peer review history of their article (what does this mean?). If published, this will include your full peer review and any attached files.

Reviewer #1: No

---

## [Author Response · Author response to Decision Letter 0]

8 Jun 2022

Dear Reviewer,

We would like to thank you for your time and efforts in evaluating our manuscript. We have reproduced your comment below, followed by a response highlighting what changes have been made in the revised manuscript.

Tobia Zanotto, PhD, Laura Rice, PhD, Jacob J Sosnoff, PhD.

Reviewer #1: Thank you for the opportunity to review this paper. It's written excellently and has robust conclusions aligned with the results and data.

My only feedback is that I find the connection between frailty and wheelchair use difficult to comprehend/ appreciate. If someone is using a wheelchair or scooter full time, isn't their chance of falling quite small- how are these patients falling over so much, is it in transition from the wheelchair/ scooter to another sitting state (e.g., to the sofa or bed?) I wonder if it's worth a mention in the intro somehow?

Reply: Thank you for the positive feedback and for this important question. Although it may be counterintuitive to think that fulltime wheelchair users may be at high risk of falls, previous research has suggested that, indeed, between 40% and 75% of individuals with MS who use wheelchairs or scooters as the primary form of mobility experience at least one fall in any six-month period (Matsuda PN, et al. Falls among adults aging with disability. Arch Phys Med Rehabil. 2015 Mar;96(3):464-71; Rice et al. Fall prevalence in people with multiple sclerosis who use wheelchairs and scooters. Medicine (Baltimore). 2017 Sep;96(35):e7860.). Particularly, the study by Rice and colleagues (2017) indicated that the majority of falls experienced by this segment of the MS population occur at home during transfers (e.g., in the bathroom). While the etiology of falls is multifactorial, it is plausible that elevated frailty levels may increase the risk of falls by impinging on the ability to transfer (e.g., through muscle weakness). We have now mentioned this additional information in the revised Introduction, as suggested. Please refer to page 4, lines 79-84 in the revised manuscript.

---

## [Editor Report · Decision Letter 1]

6 Jul 2022

Frailty among people with multiple sclerosis who are wheelchair users

PONE-D-21-38708R1

Dear Dr. Zanotto,

We’re pleased to inform you that your manuscript has been judged scientifically suitable for publication and will be formally accepted for publication once it meets all outstanding technical requirements.

Kind regards,

Sreeram V. Ramagopalan

Academic Editor

PLOS ONE
---

## [Editor Report · Acceptance letter]

8 Jul 2022

PONE-D-21-38708R1 

Frailty among people with multiple sclerosis who are wheelchair users 

Dear Dr. Zanotto:

I'm pleased to inform you that your manuscript has been deemed suitable for publication in PLOS ONE. Congratulations! Your manuscript is now with our production department. 

Kind regards, 

on behalf of

Dr. Sreeram V. Ramagopalan 

Academic Editor

PLOS ONE